# Target Identification of 22-(4-Pyridinecarbonyl) Jorunnamycin A, a Tetrahydroisoquinoline Derivative from the Sponge *Xestospongia* sp., in Mediating Non-Small-Cell Lung Cancer Cell Apoptosis

**DOI:** 10.3390/molecules27248948

**Published:** 2022-12-15

**Authors:** Iksen Iksen, Suwimon Sinsook, Onsurang Wattanathamsan, Koonchira Buaban, Supakarn Chamni, Varisa Pongrakhananon

**Affiliations:** 1Department of Pharmacology and Physiology, Faculty of Pharmaceutical Sciences, Chulalongkorn University, Bangkok 10330, Thailand; 2Pharmaceutical Sciences and Technology Program, Faculty of Pharmaceutical Sciences, Chulalongkorn University, Bangkok 10330, Thailand; 3Department of Pharmacognosy and Pharmaceutical Botany, Faculty of Pharmaceutical Sciences, Chulalongkorn University, Bangkok 10330, Thailand; 4Natural Products and Nanoparticles Research Unit (NP2), Chulalongkorn University, Bangkok 10330, Thailand; 5Preclinical Toxicity and Efficacy, Assessment of Medicines and Chemicals Research Unit, Chulalongkorn University, Bangkok 10330, Thailand

**Keywords:** 22-(4-pyridinecarbonyl) jorunnamycin A, apoptosis, mitogen-activated protein kinase, network pharmacology, non-small cell lung cancer

## Abstract

A dysregulation of the cell-death mechanism contributes to poor prognosis in lung cancer. New potent chemotherapeutic agents targeting apoptosis-deregulating molecules have been discovered. In this study, 22-(4-pyridinecarbonyl) jorunnamycin A (22-(4′py)-JA), a synthetic derivative of bistetrahydroisoquinolinequinone from the Thai blue sponge, was semisynthesized by the Steglich esterification method, and its pharmacological mechanism in non-small-cell lung cancer (NSCLC) was elucidated by a network pharmacology approach. All predicted targets of 22-(4′py)-JA and genes related to NSCLC were retrieved from drug-target and gene databases. A total of 78 core targets were identified, and their associations were analyzed by STRING and Cytoscape. Gene ontology and KEGG pathway enrichment analyses revealed that molecules in mitogen-activated protein kinase (MAPK) signaling were potential targets of 22-(4′py)-JA in the induction of NSCLC apoptosis. In silico molecular docking analysis displayed a possible interaction of ERK1/2 and MEK1 with 22-(4′py)-JA. In vitro anticancer activity showed that 22-(4′py)-JA has strong cytotoxic and apoptosis-inducing effects in H460, H292 and A549 NSCLC cells. Furthermore, immunoblotting confirmed that 22-(4′py)-JA induced apoptotic cell death in an ERK/MEK/Bcl-2-dependent manner. The present study demonstrated that 22-(4′py)-JA exhibited a potent anticancer effect that could be further developed for clinical application and showed that network pharmacology approaches are a powerful tool to illustrate the molecular pathways of new drugs or compounds.

## 1. Introduction

Lung cancer remains one of the most frequent malignancies worldwide in both sexes [1]. This aggressive pathology has the greatest mortality rate among all cancer types, accounting for over a quarter of all cancer fatalities [2]. Lung cancer is classified into non-small-cell lung cancer (NSCLC), which is a major type in approximately 85% of all cases, and small cell lung cancer (SCLC) [3]. The acquisition of chemotherapeutic resistance and high metastatic ability are contributing factors to a poor prognosis and the recurrence of tumors, both of which result in a low five-year survival rate in lung cancer patients [4]. Even though therapeutic interventions have become more advanced, most lung cancers are diagnosed at a late stage, resulting in unsatisfactory clinical outcomes [5]. The discovery of more effective drugs for lung cancer treatment is urgently needed.

Growing evidence has demonstrated that marine ecosystems are versatile resources for the discovery of novel anticancer medications [6,7]. Recent research reported that several compounds isolated from marine organisms exhibited promising anticancer activities [8,9,10,11]. Tetrahydroisoquinoline marine alkaloids, including ecteinascidins, saframycins, renieramycins, and jorunnamycins, which share a similar pentacyclic core structure along with a characteristic side chain at the C-22 position, have potent cytotoxic activities. Currently, natural and chemically modified tetrahydroisoquinolines have been approved by the US Food and Drug Administration (FDA) as anticancer drugs, such as ecteinacindin 743, for the treatment of advanced soft tissue sarcoma and ovarian cancer [12,13,14] and its derivative, lurbinectedin, for the treatment of metastatic small cell lung cancer (SCLC) [15]. Interestingly, tetrahydroisoquinolines exhibit an essential anticancer mechanism as DNA alkylating agents [16]. Renieramycin M from the Thai blue sponge *Xestospongia* sp. was able to induce apoptosis via p53 modulation, inhibit cancer metastasis, and suppress cancer stemness behaviors [17,18,19]. Recently, jorunnamycin A found in a similar blue sponge displayed intriguing antimetastatic activity and attenuated cancer stem-cell-like phenotypes [20,21], suggesting that jorunnamycin A is a potential lead compound for the development of novel anticancer agents with greater potency.

Several chemical modifications of jorunnamycin A to mimic renieramycin M and trabectedin have been reported, especially at C-22 of jorunnamycin A (Figure 1). A previous study clearly demonstrated that the leaving group of hydroxyl at C-22 is essential for cytotoxic activity [22]. Replacing this group with fluorobenzoyl and trifluoromethyl benzoyl resulted in approximately 5- to 10-fold increases in cytotoxicity against several lung cancers [23]. Currently, substitution with pyridine carbonyl at the same position results in greater potency compared to both fluorobenzoyl and trifluoromethyl benzoyl. In addition, among the derivatives of pyridine carbonyl, 4′-pyridine carbonyl ester at the C-22 position (4-pyridine carbonyl jorunnamycin A, 22-(4′py)-JA) has the strongest cytotoxicity, which is approximately 20-fold higher than those of the natural precursors renieramycin M and jorunnamycin A [23]. However, the exact mechanism of 22-(4′py)-JA remains unknown.

In the present study, 22-(4′py)-JA was semisynthesized with a slightly modified method, and its potential target and molecular signaling were identified using computational-based analysis relying on network pharmacology and molecular docking approaches (Figure 2). Furthermore, its cytotoxic effect and molecular mechanism were verified using an in vitro cell-culture-based experiment. This study provides theoretical information for the research and development of 22-(4′py)-JA for further clinical application against NSCLC.

## 2. Results

### 2.1. Semisynthesis of 22-(4′py)-JA

In the presence of 1-(3-dimethylaminopropyl)-3-ethylcarbodiimide hydrochloride (EDCI·HCl) and 4-dimethylaminopyridine (DMAP) as coupling reagents, 22-(4′py)-JA was semisynthesized from jorunnamycin A through a mild and effective Steglich esterification with isonicotinoyl chloride (Figure 3). After purification, 22-(4′py)-JA was obtained as a yellow powder at 65% yield. According to the structural elucidation via spectroscopic analysis, the characteristic chemical shifts of the additional 4′-pyridinecarbonyl motif included a pair of doublets of pyridine protons at 8.79 ppm (2′-H, 6′-H) and 7.77 ppm (3′-H, 5′-H), a carbonyl carbon at 163.0 ppm (C-24) and pyridine carbons at 147.3 ppm (C-2′, C-6′), 139.4 ppm (C-1′), and 124.1 ppm (C-3′, C-5′) (Appendix A); the 22-(4′py)-JA obtained had 88.9% purity (Appendix A).

### 2.2. Target Identification of 22-(4′py)-JA against NSCLC

Using Swiss Target Prediction, 100 targets of 22-(4′py)-JA were identified (Appendix A). According to comprehensive databases, including GeneCards, Therapeutic Target Database (TTD), Online Mendelian Inheritance in Men (OMIM), and DisGeNET, a total of 6431 disease targets related to NSCLC were obtained. Among them, a total of 78 potential targets of both 22-(4′py)-JA and NSCLC were intercepted, indicating the possible core targets of 22-(4′py)-JA against NSCLC, which is shown in a Venn diagram (Figure 4A). The compound-target interaction network was generated using Cytoscape 3.9.1 software, as shown in Figure 4B. Furthermore, 22-(4′py)-JA, an active component, is depicted as a light green hexagonal node. In addition, 78 core targets are displayed as pink oval nodes, and the connections between these nodes were made based on abundant evidence. Taken together, these results indicated that 22-(4′py)-JA has an impact on NSCLC.

### 2.3. Investigation of the Protein–Protein Interaction (PPI) Network

The PPI network was constructed from a total of 78 intercepted targets using the STRING database (Figure 5A). These core targets are shown as nodes, of which 77 nodes had an interaction. The remaining node, ADRA1A, was not connected with any other nodes. There was a total of 503 interactions among them, including the combination of neighboring genes, fusion genes, and cooccurring genes, which are represented as edges. Other parameters, including average node degree (12.9) and the average local clustering coefficient (0.589), indicated an average number of interactions of proteins and the wellness of the connected nodes in a network, respectively. A network diagram displaying the association of potential targets was constructed by importing the intersected targets into Cytoscape version 3.9.1 (Figure 5B). A total of 78 core targets were ranked according to the number of degrees, average shortest path length, betweenness centrality, closeness centrality, and clustering coefficient (Appendix A). A total of 26 targets with the top 25 highest degree values ranged from red to yellow, indicating a higher to lower degree score (Figure 5C). They included proto-oncogene tyrosine-protein kinase Src (SRC), mitogen-activated protein kinase 3 (MAPK3 or ERK1), epidermal growth factor receptor (EGFR), G1/S-specific cyclin-D1 (CCND1), mitogen-activated protein kinase 1 (MAPK1 or ERK2), mammalian target of rapamycin (MTOR), matrix metalloproteinase-9 (MMP9), receptor tyrosine-protein kinase erbB-2 (ERBB2), matrix metalloproteinase-2 (MMP2), cyclin-dependent kinase 4 (CDK4), dual-specificity mitogen-activated protein kinase 1 (MAP2K1 or MEK1), mitogen-activated protein kinase 14 (MAPK14), phosphatidylinositol 4,5-bisphosphate 3-kinase catalytic subunit alpha isoform (PIK3CA), focal adhesion kinase 1 (PTK2), histone deacetylase 1 (HDAC1), cyclin-dependent kinase 2 (CDK2), glycogen synthase kinase-3 beta (GSK3B), mitogen-activated protein kinase 8 (MAPK8), poly (ADP-ribose) polymerase 1 (PARP1), cyclin-dependent kinase 1 (CDK1), tyrosine-protein kinase JAK2 (JAK2), histone deacetylase 6 (HDAC6), insulin-like growth factor 1 receptor (IGF1R), vascular endothelial growth factor receptor 2 (KDR), matrix metalloproteinase-3 (MMP3) and prothrombin (F2), respectively.

### 2.4. Enrichment Analysis of Gene Ontology (GO) and Kyoto Encyclopedia of Genes and Genomes (KEGG) Pathways

These potential targets were subjected to three distinct types of GO functional annotation analyses. These analyses focused on biological processes, molecular functions, and cellular components. According to the degree of significance, the top ten associated functions in biological processes that participated in the targets included the response to an oxygen-containing compound, organic substance, nitrogen compound, regulation of stimulus, protein modification, phosphate metabolic process, phosphorylation, kinase activity, protein metabolic process and regulation of biological process (Figure 6A). For enrichment analysis of molecular function, these targets were involved in small-molecule binding, ribonucleotide binding, protein kinase activity, nucleotide binding, kinase activity, catalytic activity, carbohydrate derivative binding, ATP binding, anion binding and adenyl ribonucleotide binding (Figure 6B). For the cellular component, targets were mainly found in the transferase complex, serine/threonine complex, somatodendritic compartment, protein kinase complex, plasma membrane area, cytosol, cytoplasm, cyclin-dependent complex, cell junction and plasma membrane (Figure 6C).

In the pathway enrichment analysis, the top 25 pathways in which the identified targets participated are shown (Figure 6D). The pathways relevant to cancer pathogenesis were proteoglycans in cancer, prostate cancer, prolactin signaling, PI3K-AKT signaling, pathway in cancer, pancreatic cancer, microRNA in cancer, growth hormone, FoxO signaling, focal adhesion, ErbB signaling, EGFR tyrosine kinase inhibitor resistance, colorectal cancer, autophagy, T-cell receptor signaling, sphingolipid signaling, and apoptosis. A list of the targets associated with these pathways is listed in Table 1.

### 2.5. Potential Targets and Signaling Pathway of 22-(4′py)-JA in Regulating Cancer Apoptosis

An increasing number of studies have reported that the aberrant regulation of apoptosis genes plays significant roles in cancer progression and therapeutic resistance, and apoptosis sensitization is a major therapeutic strategy for anticancer drug research and discovery [24]. The 14 potential targets of 22-(4′py)-JA in regulating NSCLC apoptosis are shown in Table 1, of which the top three targets ranked by score of the degree were MAPK3 (ERK1), MAPK1 (ERK2) and MAP2K1 (MEK1) (Appendix A). The association of these targets in the apoptosis pathway and their signaling relevance were constructed using the KEGG pathway (Appendix A). Based on our findings, candidate targets of 22-(4′py)-JA might participate in MAPK/Bcl-2 signaling in mediating NSCLC apoptosis.

### 2.6. Molecular Docking and Molecular Dynamic Analysis of 22-(4′py)-JA-Target Interactions

Interactions of 22-(4′py)-JA and three candidate targets associated with the apoptosis pathway were investigated. Molecular docking demonstrated binding between 22-(4′py)-JA and either ERK1, ERK2 or MEK1 by hydrogen bonding and van der Waals, hydrophobic and electrostatic interactions (Figure 7A–C). The results showed that 22-(4′py)-JA interacted with the kinase domains of ERK1 (42–330 amino acids), ERK2 (25–313 amino acids) and MEK1 (75-393 amino acids), whose binding energies were −9.9, −8.8 and −9.1 kcal/mol, respectively, and their ligand efficiencies were −0.225, −0.2 and −0.207 kcal/mol per heavy atom, respectively (Table 2).

Furthermore, molecular dynamics simulations between 22-(4′py)-JA and these targets were performed. As shown in Figure 7D, the average root mean square deviation (RMSD) values of the ligand conformations for ERK1, ERK2 and MEK1 were 1.75 ± 0.59, 1.3 ± 0.14 and 1.42 ± 0.18 Å, respectively. The interaction between 22-(4′py)-JA and ERK1 underwent sharp fluctuation in the first 6 ns and remained stable until the end of the experiment, while the interaction between 22-(4′py)-JA and either ERK2 or MEK1 maintained a closer proximity throughout 15 ns. In addition, the RMSD of ligand movement between 22-(4′py)-JA and these targets was determined and showed that the average RMSD values were 6.11 ± 2.14, 2.44 ± 0.36 and 4.16 ± 2.17 Å for ERK1, ERK2 and MEK1, respectively (Figure 7E). A lower RMSD of ligand movement indicated a closer proximity of targets, and it was clear that the interaction between 22-(4′py)-JA with either ERK1 or MEK1 had a higher tendency to fluctuate than that for ERK2. Taken together, these data suggested that 22-(4′py)-JA has multiple targets in the MAPK family, and ERK2 showed the most potential and stable interaction with 22-(4′py)-JA.

### 2.7. In Vitro Target Validation of 22-(4′py)-JA in Apoptosis-Inducing Effects

To validate the candidate target identified, the anticancer activity of 22-(4′py)-JA was first investigated in NSCLC cells named H460, A549 and H292. Cytotoxicity was examined by MTT assays, and the results demonstrated that 22-(4′py)-JA significantly reduced the number of viable cells. The 50% inhibitory concentrations (IC_50_) were 18.9 ± 0.76, 14.43 ± 0.68, and 16.95 ± 0.41 nM in H460, A549 and H292 cells, respectively (Figure 8A). However, it has a less cytotoxic effect on normal lung epithelial BEAS-2B cells with IC_50_ ≥ 100 nM (Appendix A), suggesting that this compound induces cell death preferentially in lung cancer cells. The apoptosis-inducing effect of 22-(4′py)-JA was then assessed by Hoechst 33342 and annexin-V/propidium iodide (PI) staining assays. Treatment with 22-(4′py)-JA obviously increased the number of cells with apoptotic nuclei, chromatin condensation and apoptotic body formation (Figure 8B). The percentages of apoptosis were approximately 50, 65 and 80% in all tested cells in response to 20, 40 and 80 nM 22-(4′py)-JA, respectively. Furthermore, 22-(4′py)-JA also significantly induced an increase in early apoptotic (Annexin-V^+^, PI^−^) and late apoptotic (Annexin-V^+^, PI^+^) cells in a dose-dependent manner (Figure 8C). In addition, the ratio of cleaved-caspase 3 to total caspase-3 was upregulated in response to 22-(4′py)-JA, while there was no effect in JA-treated cells with similar concentrations (Appendix A). These data indicated that 22-(4′py)-JA exhibited a potent cytotoxic effect and strongly mediated cell death via an apoptosis mechanism.

To verify the molecular mechanism by which 22-(4′py)-JA mediates NSCLC apoptosis, NSCLC cells were treated with various concentrations of 22-(4′py)-JA, and protein expression was investigated by immunoblotting. The results revealed that 22-(4′py)-JA markedly downregulated phosphorylated MEK1/2 (p-MEK1/2) and p-ERK1/2 and consequently decreased antiapoptotic Bcl-2 levels (Figure 9A,B). However, the total forms of MEK1/2 and ERK1/2 were not altered. Since EGF, a ligand of receptor tyrosine kinase, is widely reported to promote tumor growth, cell proliferation and apoptosis deregulation through activation of its downstream signaling, including MEK/ERK [25], we clarified whether 22-(4′py)-JA was able to suppress MEK/ERK signaling mediated by EGF. Supporting our finding, 22-(4′py)-JA remarkedly decreased a number of viable cells (Figure 9C). Since 22-(4′py)-JA exhibited the highest cytotoxicity on A549 cells (Figure 8), the effect of 22-(4′py)-JA on EGF regulating these signaling was performed in this cell type. The results showed that 22-(4′py)-JA remarkably suppressed an increase in MEK and ERK phosphorylation and Bcl-2 expressions that were mediated by EGF (Figure 9D). Taken together, these data indicated that MAPK/Bcl-2 signaling is a mode of action of 22-(4′py)-JA in apoptosis induction in NSCLC.

## 3. Discussion

Lung cancer is an aggressive type of cancer with a high mortality rate, and the discovery of novel anticancer drugs is challenging. Several anticancer drugs approved by the FDA were originally from natural sources, such as paclitaxel from *Taxus brevifolia* [26] and trabectedin from *Ecteinascidia turbinate* [13]. A previous study reported that JA, isolated from a Thai blue sponge, had cytotoxic effects in a variety of cancer cell lines, including NSCLC, colon cancer, and breast cancer cell lines [22,23]. To enhance the potency, 22-(4′py)-JA was semisynthesized using JA as a precursor. Furthermore, this compound exhibited stronger cytotoxicity in lung cancer, with a potency 15- to 20-fold greater than JA [23]. In a tiered approach using in silico network pharmacology and molecular docking, targets of 22-(4′py)-JA against NSCLC were first identified, and it was validated that ERK1/2 and MEK1 played potential key roles in the pharmacological mechanism of anti-lung cancer activity.

The identification of the pharmacological mechanisms of new drugs is important. Several approaches have been proposed, such as in vitro high-throughput screening and gene/protein array-based assays [27,28]; however, their time-cost effectiveness can be an impediment [29]. An in silico network pharmacology strategy integrating molecular docking has arisen that provides a better understanding of how drugs interact with multiple targets [30] and explains the complex connections between natural compounds and human biological molecules [31]. Due to an intriguing algorithm and the accessibility of data mining, this approach is a widely acceptable and powerful tool for predicting potential targets that is more precise and less time-consuming and thus minimizes the expense of drug discovery, facilitating an increase in the success rate of clinical trials [31,32].

Based on the analysis of the protein–protein interaction network, the top 25 targets from 22-(4′py)-JA were related to NSCLC aggressiveness and found to participate in cancer-associated pathways. Apoptosis is a major mode of action of cancer therapies, and resistance to this programmed cell death facilitates long-lived cancer cells with an accumulation of gene mutations, contributing to uncontrolled cell growth, vascularization and tumor invasion [24,33]. We then focused on the apoptosis targets of 22-(4′py)-JA in NSCLC. According to our findings, the potential targets of 22-(4′py)-JA were mainly members of the MAPK family. The MAPK superfamily consists of protein kinases that function in signal transduction by phosphorylating their downstream signaling pathways [34]. There are three main subtypes of MAPK, including ERK, JNK and p38 mitogen-activated protein kinases (p38). Each MAPK group is stimulated by several growth factors associated with their receptors or external stimuli, but it has a distinct, specific substrate. In the ERK cascade, upon stimulation by mitogens or growth factors, growth receptors recruit growth factor receptor-binding protein 2 (Grb2) and the C-terminus of son of sevenless (SOS) to form a complex that is required for Ras/Raf activation. The catalytic domain of Raf binds to and activates MEK, which induces ERK phosphorylation at serine-threonine residues, leading to the sequential stimulation of downstream signaling cascades of cancer-related genes. This signal transduction is extensively implicated in tumorigenesis, malignancy development and chemotherapeutic resistance [35,36]. Accumulative studies have reported that overactivation of MAPK signaling leads to the suppression of various proapoptotic factors, such as BAX, caspase 3 and poly (ADP-ribose) polymerase (PARP), in contrast to the upregulation of the antiapoptotic protein Bcl-2 [34,37,38]. Moreover, an attenuation of ERK activity was shown to trigger apoptotic cell death [38,39]. Likewise, MEK1 was reported as a prognostic marker, and its chemical inhibitor was able to sensitize apoptosis [40]. A MEK1 inhibitor is undergoing preclinical evaluation for several cancers [41,42], indicating that the suppression of these molecules is a promising therapeutic approach for cancer treatment.

According to their molecular structures, ERK1, ERK2 and MEK1 share similar homology, consisting of an N-terminal domain, a kinase domain, an activation loop, a TEY motif and a C-terminal domain. An in silico study demonstrated that 22-(4′py)-JA could interact with ERK1, ERK2 and MEK1, especially at the kinase domain. All of the interactions showed the great potency of 22-(4′py)-JA as an inhibitor of the MAPK family, for which the binding energy was lower than -5 Kcal/mol [43]. Based on our findings, in silico and in vitro preclinical evaluations verified that ERK1/2 and MEK1 are molecular targets of 22-(4′py)-JA in inducing NSCLC apoptosis. Additionally, 22-(4′py)-JA suppressed these target activities, thereby decreasing anti-apoptotic Bcl-2 levels and mediating cell death via an apoptosis mechanism.

From the point of view of network pharmacology, this study explains the putative active compound, possible targets and essential biological pathways of 22-(4′py)-JA in the treatment of NSCLC, especially by inducing apoptosis. Further study on the analysis of structure-activity relationship of derivatives will be carried out to obtain more potent compounds with safer toxicity profiles. This study, at least, provides an intriguing approach for the target identification of new compounds and scientific information for further research and development of this compound for clinical application against NSCLC.

## 4. Materials and Methods

### 4.1. Chemicals and Reagents

Hoechst 33342 and 3-(4,5-dimethylthiazol-2-yl)-2,5-diphenyltetrazolium bromide (MTT) were purchased from Invitrogen (MA, USA). Dimethyl sulfoxide (DMSO) and epidermal growth factor (EGF) were purchased from Sigma-Aldrich (MI, USA). Chloroform (CHCl_3_), deuterated chloroform (CDCl_3_), dichloromethane (CH_2_Cl_2_), dimethyl sulfoxide, ethyl acetate (CH_3_COOC_2_H_5_), hexane (C_6_H_14_), isotonicotinoyl chloride hydrochloride (C_6_H_4_ClNO), 1-(3-dimethylaminopropyl)-3-ethylcarbodiimide hydrochloride (C_8_H_17_N_3_·HCl) (ECDI), dichloromethane (CH_2_Cl_2_) and silica gel 7734 were purchased from Tokyo Chemical Industry (Tokyo, Japan).

### 4.2. Semisynthesis of 22-(4′py)-JA from Jorunnamycin A

Jorunnamycin A was isolated from the Thai blue sponge, *Xestospongia* sp., which was collected by SCUBA diving in the vicinity of Si-Chang Island, Chonburi Province, Thailand [21]. Solvents, which included ethyl acetate, dichloromethane, and hexane, were distilled before use. Jorunnamycin A (25.00 mg, 0.05066 mmol, 1 equiv.) was weighed into an oven-dried round-bottomed flask and dissolved in dichloromethane (CH_2_Cl_2_) (10 mL). Following this, 4-dimethylaminopyridine (DMAP) (30.94 mg, 0.2532 mmol, 5 equiv.), 1-(3-dimethylaminopropyl)-3-ethylcarbodiimide hydrochloride (EDCI·HCl) (48.56 mg, 0.2532 mmol, 5 equiv.) and isonicotinoyl chloride (45.08 mg, 0.2532 mmol, 5 equiv.) were sequentially added to the reaction. The reaction mixture was magnetically stirred with a nitrogen (N_2_) gas balloon in an inert atmosphere for 1 h at room temperature. The reaction was monitored by thin-layer chromatography (TLC) using a mixture solvent of ethyl acetate: hexane (1:1) as a mobile phase. After TLC showed completion, the reaction was concentrated under reduced pressure. The resulting crude product was purified by flash column chromatography using silica gel as the stationary phase and a mixed solvent of ethyl acetate and hexane as the mobile phase to obtain 16.2 mg (65% yield) of 22-(4′py)-JA as a yellow amorphous powder. The desired product was characterized by spectroscopic methods. The ^1^H and ^13^C nuclear magnetic resonance (NMR) spectra were obtained on a Bruker ADVANCE NEO 400 MHz NMR spectrometer. Deuterated chloroform (CDCl_3_) served as the internal standard for both the 1H (7.26 ppm) and 13C (77.0 ppm) NMR spectra. Accurate mass spectra were obtained with an Agilent 6540 UHD Q-TOF LC/MS spectrometer. The spectroscopic data corresponded with a previous report [22,23].

### 4.3. Target Identification of 22-(4′py)-JA and NSCLC-Related Genes

In silico high-throughput screening of NSCLC therapeutic targets was obtained from GeneCards [44], Therapeutic Targets Database (TTD) [45], Online Mendelian Inheritance in Man (OMIM) [46] and DisGeNET [47]. The possible targets of the compounds were retrieved from the Swiss Target Prediction database (http://www.swisstargetprediction.ch, accessed on 14 September 2022) [48]. The intercept targets between 22-(4′py)-JA and NSCLC are presented as a Venn diagram, which was constructed by Venny 2.1.0 [49].

### 4.4. Compound–Target and Protein–Protein Network Construction

The potential targets of 22-(4′py)-JA were constructed by Cytoscape 3.9.1 [50]. The protein–protein interaction network was obtained from STRING v11.5 [51] by the addition of the intersection targets of 22-(4′py)-JA and NSCLC. The protein type was set to “Homo sapiens,” with a 0.4 level of confidence, and the other parameters were set to default values. The protein interacting relationships were retrieved and imported into Cytoscape 3.9.1, and an interaction network was constructed. The analysis of the top 26 targets was conducted using the cytoHubba plugin [52]. The core proteins with the top 25 highest degree values were subjected to subsequent analyses.

### 4.5. Bioinformatic Analysis of Gene Ontology (GO) and Kyoto Encyclopedia of Genes and Genomes (KEGG) Pathways

Both bioinformatic analyses were retrieved from the STRING database by importing the intercept targets of 22-(4′py)-JA and NSCLC. The functionality of genes, including biological processes, cellular components and molecular functions, was examined by GO analysis. The putative molecular mechanism of 22-(4′py)-JA in NSCLC was performed by KEGG pathway enrichment analysis [53], and the apoptosis pathway (hsa04210) was generated. The data were presented as a bubble plot with a ggplot2 script using Rstudio software [54].

### 4.6. Molecular Docking and Dynamics

The X-ray crystal structures of ERK1, ERK2 and MEK1 were retrieved from the Protein Data Bank (PDB) with PDB IDs of 6GES, 1WZY and 7PQV, respectively. The structure of 22-(4′py)-JA was drawn by ChemDraw Ultra 15.0 (Perkin Elmer, MA, USA). The molecular docking of 22-(4′py)-JA and protein targets was performed by the PyRx Virtual Screening Tool (Version 0.8). The ligand conformations with the highest clusters were analyzed for free binding energies (ΔG). The binding interactions between ligands and target proteins were analyzed using PyMOL (Schrödinger, OR, USA) and BIOVIA Discovery Studio Visualizer 2022 (Biovia, CA, USA). The molecular dynamics simulation was performed by Yasara software with the AMBER14 force field implemented in the YASARA structure. The simulation was carried out at 298 K and pH 7.4 for 15 ns. The analysis was performed by employing the default macro md_run.mcr and md_analyse.mcr. The RMSD graphic was generated using Rstudio software [54].

### 4.7. Cell Culture

Human NSCLC H460, H292 and A549 cells were obtained from the American Type Culture Collection (ATCC, VA, USA). H460 and H292 cells were cultured in Roswell Park Memorial Institute (RPMI) 1640 medium, while A549 cells were cultured in Dulbecco’s modified Eagle’s medium (DMEM). All media were supplemented with 10% fetal bovine serum (FBS), 100 U/mL penicillin–streptomycin antibiotic solution and 2 mM L-glutamine. Cells were maintained in a 37 °C humidified incubator with 5% CO_2_. All media and supplements were purchased from GIBCO (MA, USA).

### 4.8. Cytotoxicity Testing

The MTT colorimetric assay was used to evaluate cell viability. H460, H292 and A549 cells were seeded onto 96-well plates at a density of 5 × 10^3^ cells per well before being incubated in an incubator at 37 °C overnight. Cells were treated with varying concentrations of 22-(4′py)-JA, ranging from 0 to 100 nM for 48 h. Subsequently, 100 µL of MTT solution (0.5 mg/mL) was added, and cells were incubated for another 4 h. The formazan crystal was dissolved in DMSO, and the optical intensity was measured by a microplate reader (Perkin Elmer VICTOR3/Wallac 1420, MA, USA) at 570 nm. Cell viability was calculated as a percentage, and the inhibitory concentration at 50% (IC_50_) was analyzed by Prism 9 (GraphPad Software, CA, USA).

### 4.9. Apoptosis Assay

Apoptosis cells were evaluated by Hoechst 33342 and annexin-V/propidium iodide (PI) staining assays. For Hoechst 33342 staining, H460, H292 and A549 cells were seeded at a density of 5 × 10^3^ cells/well in 96-well plates and treated with varying doses of 22-(4′py)-JA for 48 h. The cells were incubated with 10 µM Hoechst 33342 for 15 min in the dark at 37 °C. Apoptotic nuclei were visualized and imaged in at least 5 random fields/well by a fluorescence microscope (ECLIPSE Ts2; Nikon, New York, NY, USA). The number of apoptotic cells was calculated and presented as a percentage.

For apoptosis detection by Annexin-V/PI staining, it was performed as previously described with slight modification [37]. Cells were treated with 22-(4′py)-JA for 48 h; apoptosis cells were determined using an apoptosis detection kit (Invitrogen, MA, USA). Briefly, cells were washed with cold PBS and resuspended in a binding buffer. Cells were then incubated with Annexin-V-FITC/propidium iodide (PI) solution for 15 min at room temperature. The dye intensity in each single cell was analyzed by EPICS-XL flow cytometer (Beckman Coulter, IN, USA).

### 4.10. Immunoblotting Analysis

H460, H292 and A549 cells were seeded at a density of 3 × 10^5^ cells/well in 6-well plates and treated with 22-(4′py)-JA for 24 h. Cells were lysed with TMN lysis buffer (20 mM Tris-HCl, 1 mM MgCl_2_, 150 mM NaCl; 20 mM NaF, 0.5% sodium deoxycholate, 1% nonidet-40, 0.1 mM phenylmethylsulfonyl fluoride) with protease inhibitor cocktail (Roche Diagnostics, IN, USA) as previously reported [55]. Total proteins were separated by SDS–PAGE and electrotransferred to a PVDF membrane (CA, USA). The membrane was incubated with specific primary antibody at 4 °C overnight and secondary antibody for 2 h at room temperature. The antibodies used in this study were anti-p-ERK1/2 (cat. No. 4376), anti-ERK (cat. No. 4695), anti-p-MEK1/2 (cat. No. 9154), anti-MEK1/2 (cat. No. 8727), anti-Bcl-2 (cat. No. 4223), anti-caspase 3 (cat. No. 9665), anti-GAPDH (cat. No. 97166), anti-rabbit-HRP (cat. No. 7074), and anti-mouse HRP (cat. No. 7076). These antibodies were obtained from Cell Signaling Technology (MA, USA). Protein expression was examined by an enhanced chemiluminescence system using either SuperSignal West Pico (Thermo Fisher Scientific, MA, USA) or Immobilon Western (EMD Millipore, Darmstadt, Germany) and quantified using ImageJ software (NIH).

### 4.11. Statistical Analysis

The data are presented as the mean ± standard deviation from at least three independent experiments. The statistical analysis of all data was performed by GraphPad Prism 9 (GraphPad Software, CA, USA). One-way analysis of variance (ANOVA) followed by Tukey’s multiple comparison test was used to assess statistical significance with a *p* value < 0.05.

## 5. Conclusions

In the present study, the targets of 22-(4′py)-JA in NSCLC were first identified by a network pharmacology strategy. The results revealed 78 potential targets of 22-(4′py)-JA against NSCLC. Aberrant apoptotic cell death provides an unsatisfactory clinical outcome, and the underlying mechanisms are of great interest for new drug discovery. Our findings demonstrated that 22-(4′py)-JA exhibited a potent apoptosis-inducing effect. A cell-based assay was integrated with bioinformatic analysis, and among the predicted targets, ERK1/2 and MEK1 were identified as molecular targets of 22-(4′py)-JA in mediating NSCLC apoptosis. This study provides an intriguing approach for target identification of new compounds and scientific information for further research and development of this compound for clinical application against NSCLC.

## Figures and Tables

**Figure 1 molecules-27-08948-f001:**
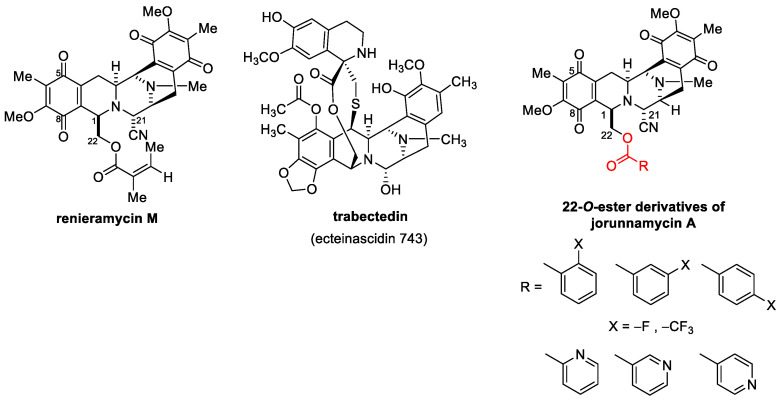
22-*O*-Ester Derivatives of jorunnamycin A [23].

**Figure 2 molecules-27-08948-f002:**
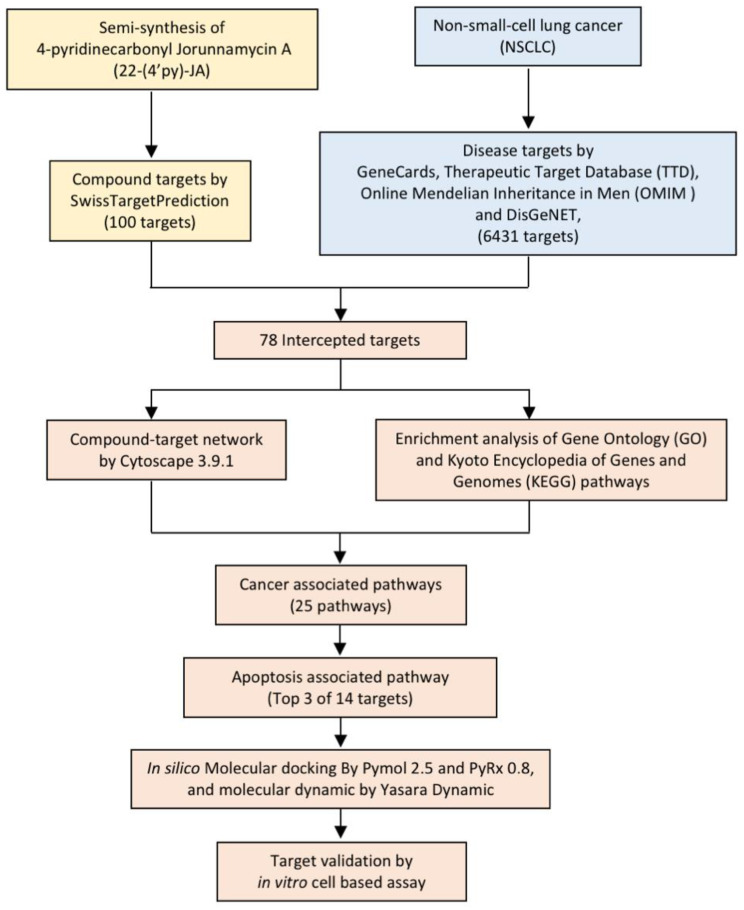
Experimental design.

**Figure 3 molecules-27-08948-f003:**
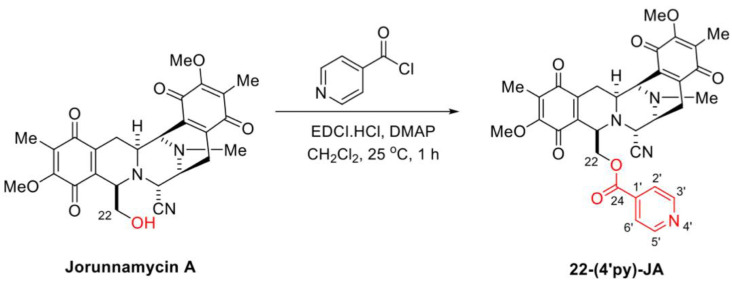
Semisynthesis of 22-(4′py)-JA.

**Figure 4 molecules-27-08948-f004:**
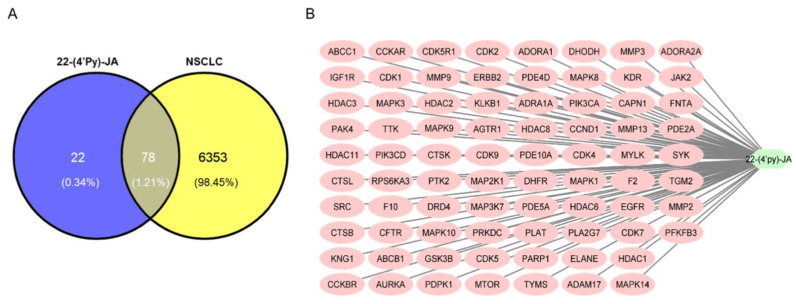
Compound target-disease target identification. (**A**) Venn diagrams represent the target interception between compound and disease. The blue part represents the number of 22-(4′py)-JA targets, and the yellow part represents the number of non-small-cell lung cancer (NSCLC) targets. (**B**) A compound-target network was constructed by Cytoscape 3.9.1.

**Figure 5 molecules-27-08948-f005:**
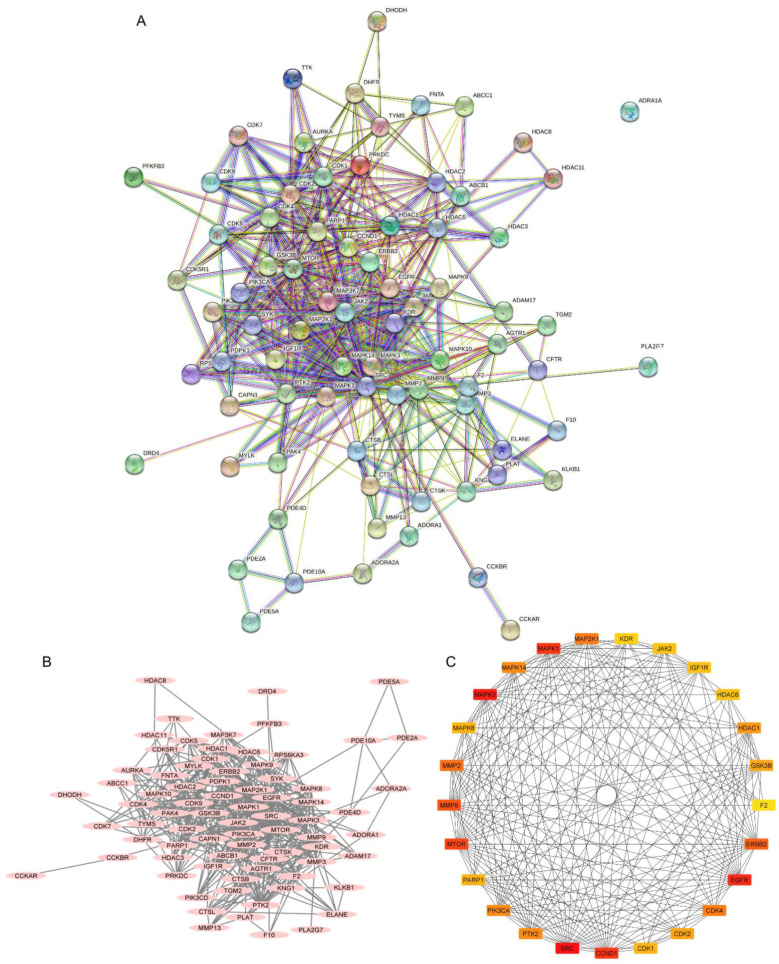
Protein–protein interactions (PPIs) of non-small-cell lung cancer (NSCLC) targets related to 22-(4′py)-JA. (**A**) The PPI network was constructed by importing the 78 overlapping targets and analyzing their interactions using the search tool from the STRING database. (**B**) The association of proteins was constructed by Cytoscape 3.9.1. Proteins are represented by nodes with pink ovals, while the edges indicate the association between proteins. (**C**) The top 26 core targets with the top 25 highest degree scores were generated by the cytoHubba plug-in. Colors ranging from red to yellow indicate a higher to lower score of degree.

**Figure 6 molecules-27-08948-f006:**
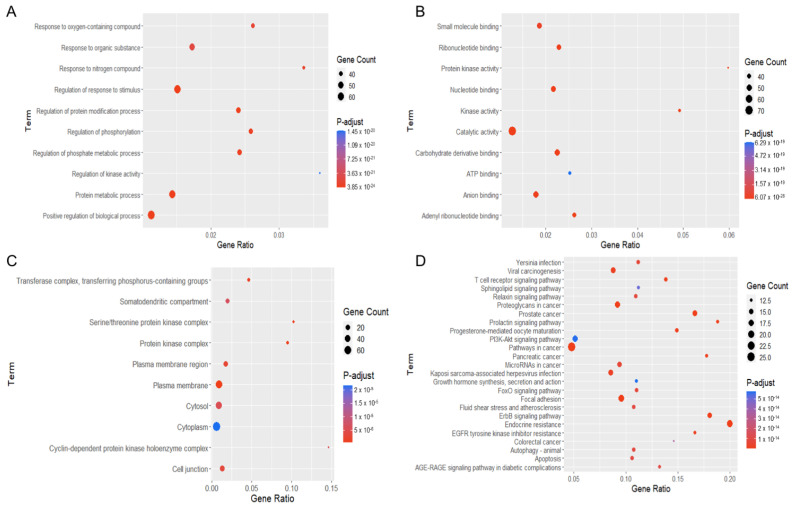
Gene ontology (GO) and Kyoto Encyclopedia of Genes and Genomes (KEGG) pathway enrichment analyses of potential targets of 22-(4′py)-JA against non-small-cell lung cancer (NSCLC). The encoded data obtained from the STRING database were constructed as a graphic using RStudio with the ggplot2 plugin. The GO terms consist of (**A**) biological process, (**B**) molecular function and (**C**) cellular component. (**D**) The KEGG pathway of potential target genes of 22-(4′py)-JA in NSCLC was analyzed.

**Figure 7 molecules-27-08948-f007:**
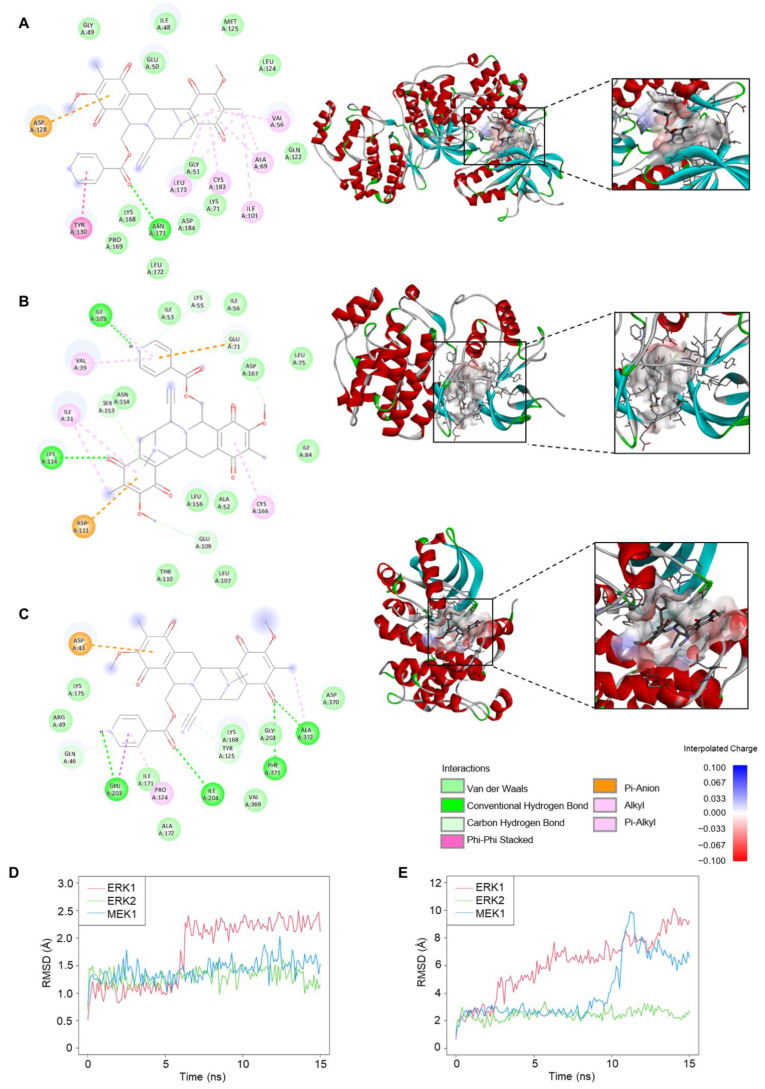
Molecular docking and molecular dynamics between 22-(4′py)-JA and protein targets. The 2D and 3D interactions between 22-(4′py)-JA and ERK1 (**A**), ERK2 (**B**) and MEK1 (**C**). RMSD for ligand conformation (**D**) and ligand movement (**E**). A simulation between 22-(4′py)-JA and ERK1 (red), ERK2 (green) and MEK1 (blue) for 15 ns were plotted.

**Figure 8 molecules-27-08948-f008:**
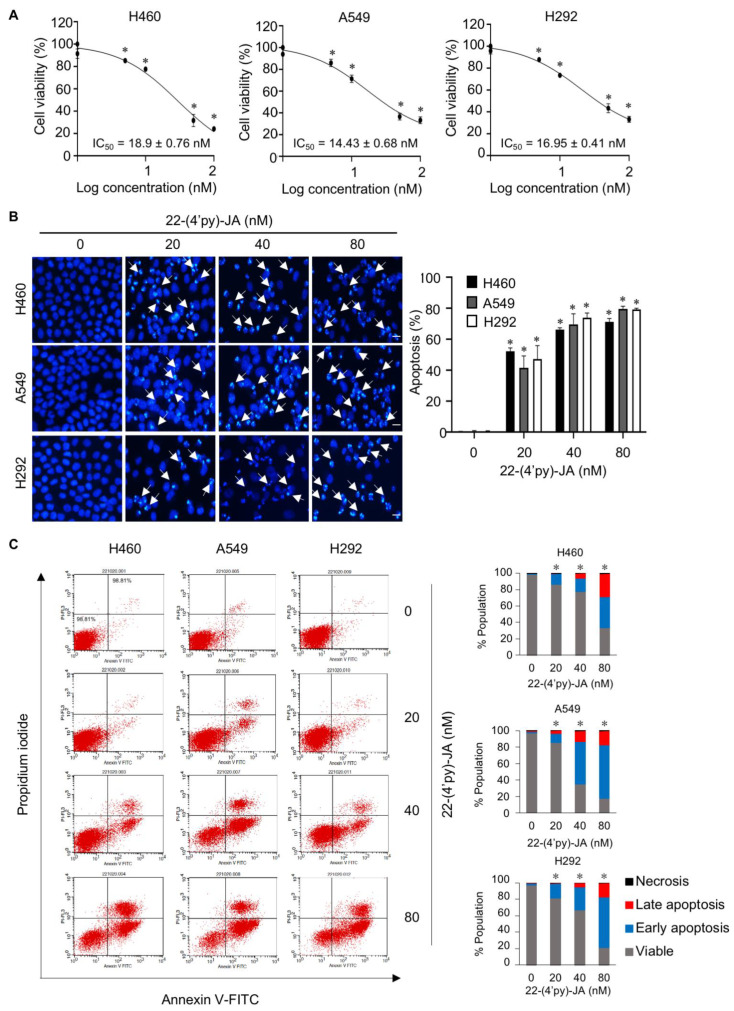
In vitro target validation of 22-(4′py)-JA and its apoptosis-inducing effect. H460, A549 and H292 cells were incubated with 22-(4′py)-JA for 48 h. (**A**) Cytotoxic effects of 22-(4′py)-JA were evaluated by MTT assay, represented as a percentage of cell viability. (**B**) Nuclei were stained with Hoechst 33342 and captured using a fluorescence microscope. Apoptotic nuclei are indicated by arrows and calculated as the percentage of apoptotic cells. Scale bar is 10 μm. (**C**) Apoptosis cells were evaluated by Annexin-V/propidium iodide (PI) staining. Histogram demonstrated the population of cells that were early apoptotic (Annexin-V^+^, PI^−^), late apoptotic (Annexin-V^+^, PI^+^), necrotic (Annexin-V^−^, PI^+^) and viable (Annexin-V^−^, PI^−^). Data are the mean ± SD (*n* = 3). * *p* < 0.05 vs. untreated control cells.

**Figure 9 molecules-27-08948-f009:**
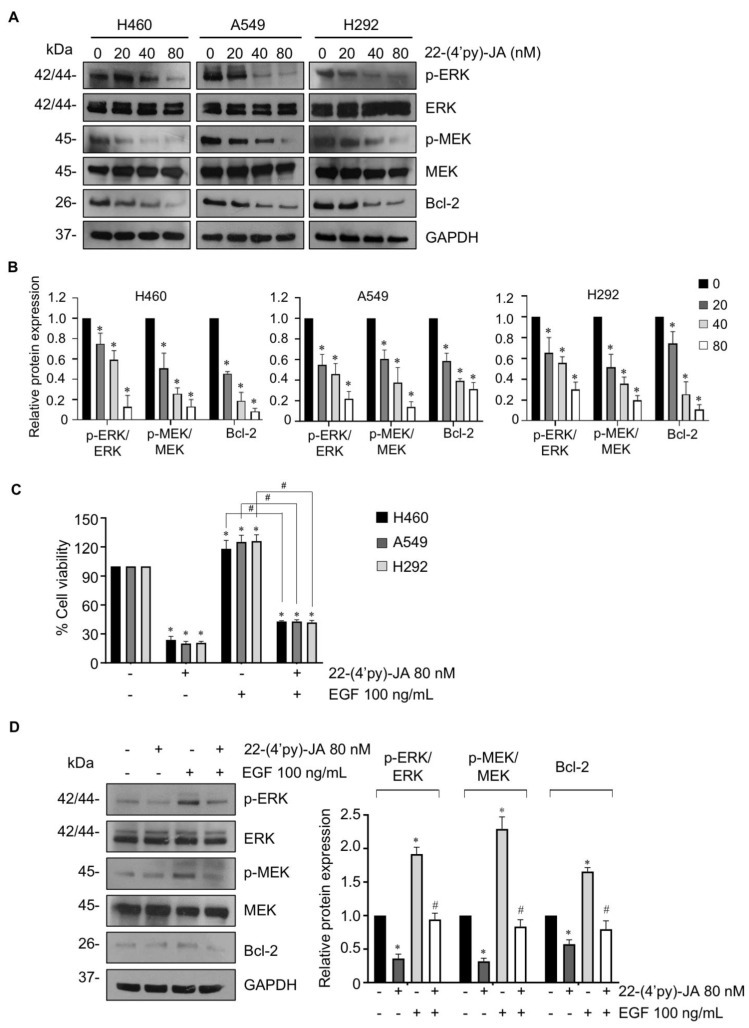
The effect of 22-(4′py)-JA on MEK/ERK/Bcl-2 expressions. (**A**) H460, A549 and H292 cells were incubated with 22-(4′py)-JA for 24 h. The expression levels of phosphorylated MEK1/2 (p-MEK1/2), MEK1/2, p-ERK1/2, ERK1/2 and Bcl-2 were analyzed by immunoblotting. Blots were reprobed with anti-GAPDH as a loading control. Representative blots from triplicate independent experiments are shown. (**B**) Densitometry analysis of protein expression was performed and presented as a relative value to the control group. Data are the mean ± SD (*n* = 3). * *p*  <  0.05 vs. untreated control cells. (**C**) H460, A549 and H292 cells were pretreated with epidermal growth factor (EGF) for 1 h, prior to 22-(4′py)-JA (80 nM) for 48 h. Cell viability was examined by MTT assay and represented as a percentage. (**D**) A549 cells were pretreated with epidermal growth factor (EGF) for 1 h, prior to 22-(4′py)-JA (80 nM) for 24 h. The expression levels of phosphorylated MEK1/2 (p-MEK1/2), MEK1/2, p-ERK1/2, ERK1/2 and Bcl-2 were analyzed by immunoblotting. Blots were reprobed with anti-GAPDH as a loading control. Densitometry analysis of protein expression was performed and presented as a relative value to the control group. Data are the mean ± SD (n = 3). * *p*  < 0.05 vs. untreated control cells. # *p* < 0.05 vs. EGF-treated cells.

**Table 1 molecules-27-08948-t001:** Pathways related to 22-(4′py)-JA against NSCLC.

Pathways	Targets
AGE-RAGE signaling pathway in diabetic complications	MAPK1, MMP2, CCND1, MAPK14, CDK4, MAPK3, PIK3CA, MAPK10, PIK3CD, JAK2, MAPK8, MAPK9, AGTR1
Apoptosis	MAPK1, MAPK3, PIK3CA, CTSK, MAP2K1, PDPK1, CTSL, CTSB, MAPK10, PARP1, PIK3CD, MAPK8, MAPK9, CAPN1
Autophagy–animal	MAPK1, MAPK3, PIK3CA, IGF1R, MAP2K1, PDPK1, CTSL, CTSB, MAPK10, MTOR, MAP3K7, PIK3CD, MAPK8, MAPK9
Colorectal cancer	MAPK1, CCND1, MAPK3, PIK3CA, EGFR, MAP2K1, GSK3B, MAPK10, MTOR, PIK3CD, MAPK8, MAPK9
EGFR tyrosine kinase inhibitor resistance	MAPK1, MAPK3, KDR, PIK3CA, IGF1R, ERBB2, EGFR, MAP2K1, GSK3B, MTOR, SRC, PIK3CD, JAK2
Endocrine resistance	MAPK1, MMP2, CCND1, MAPK14, CDK4, MAPK3, PIK3CA, IGF1R, ERBB2, EGFR, MAP2K1, PTK2, MAPK10, MTOR, MMP9, SRC, PIK3CD, MAPK8, MAPK9
ErbB signaling pathway	MAPK1, MAPK3, PIK3CA, ERBB2, EGFR, MAP2K1, GSK3B, PTK2, MAPK10, MTOR, SRC, PIK3CD, MAPK8, MAPK9, PAK4
Fluid shear stress and atherosclerosis	MMP2, PLAT, MAPK14, KDR, PIK3CA, PTK2, CTSL, MAPK10, MAP3K7, MMP9, SRC, PIK3CD, MAPK8, MAPK9
Focal adhesion	MAPK1, CCND1, MAPK3, KDR, PIK3CA, IGF1R, ERBB2, EGFR, MAP2K1, GSK3B, PTK2, PDPK1, MAPK10, MYLK, SRC, PIK3CD, MAPK8, MAPK9, PAK4
FoxO signaling pathway	MAPK1, CCND1, MAPK14, MAPK3, PIK3CA, CDK2, IGF1R, EGFR, MAP2K1, PDPK1, MAPK10, PIK3CD, MAPK8, MAPK9
Growth hormone synthesis, secretion and action	MAPK1, MAPK14, MAPK3, PIK3CA, MAP2K1, GSK3B, PTK2, MAPK10, MTOR, PIK3CD, JAK2, MAPK8, MAPK9
Kaposi sarcoma-associated herpesvirus infection	MAPK1, CCND1, MAPK14, CDK4, MAPK3, PIK3CA, MAP2K1, GSK3B, MAPK10, MTOR, SRC, SYK, PIK3CD, JAK2, MAPK8, MAPK9
MicroRNAs in cancer	MAPK1, CCND1, MAPK3, PIK3CA, ERBB2, EGFR, MAP2K1, MTOR, MMP9, HDAC1, PIK3CD, ABCC1, HDAC2, PAK4, ABCB1
Pancreatic cancer	MAPK1, CCND1, CDK4, MAPK3, PIK3CA, ERBB2, EGFR, MAP2K1, MAPK10, MTOR, PIK3CD, MAPK8, MAPK9
Pathways in cancer	MAPK1, MMP2, CCND1, CDK4, MAPK3, PIK3CA, KNG1, CDK2, IGF1R, ERBB2, EGFR, MAP2K1, F2, GSK3B, PTK2, MAPK10, MTOR, MMP9, HDAC1, PIK3CD, JAK2, MAPK8, MAPK9, AGTR1, HDAC2
PI3K-Akt signaling pathway	MAPK1, CCND1, CDK4, MAPK3, KDR, PIK3CA, CDK2, IGF1R, ERBB2, EGFR, MAP2K1, GSK3B, PTK2, PDPK1, MTOR, SYK, PIK3CD, JAK2
Progesterone-mediated oocyte maturation	MAPK1, AURKA, MAPK14, MAPK3, PIK3CA, CDK2, IGF1R, MAP2K1, MAPK10, PIK3CD, RPS6KA3, CDK1, MAPK8, MAPK9
Prolactin signaling pathway	MAPK1, CCND1, MAPK14, MAPK3, PIK3CA, MAP2K1, GSK3B, MAPK10, SRC, PIK3CD, JAK2, MAPK8, MAPK9
Prostate cancer	MAPK1, PLAT, CCND1, MAPK3, PIK3CA, CDK2, IGF1R, ERBB2, EGFR, MMP3, MAP2K1, GSK3B, PDPK1, MTOR, MMP9, PIK3CD
Proteoglycans in cancer	MAPK1, MMP2, CCND1, MAPK14, MAPK3, KDR, PIK3CA, IGF1R, ERBB2, EGFR, MAP2K1, PTK2, PDPK1, CTSL, MTOR, MMP9, SRC, PIK3CD
Relaxin signaling pathway	MAPK1, MMP2, MAPK14, MMP13, MAPK3, PIK3CA, EGFR, MAP2K1, MAPK10, MMP9, SRC, PIK3CD, MAPK8, MAPK9
Sphingolipid signaling pathway	MAPK1, MAPK14, MAPK3, PIK3CA, KNG1, MAP2K1, PDPK1, MAPK10, ADORA1, PIK3CD, MAPK8, ABCC1, MAPK9
T-cell receptor signaling pathway	MAPK1, MAPK14, CDK4, MAPK3, PIK3CA, MAP2K1, GSK3B, PDPK1, MAPK10, MAP3K7, PIK3CD, MAPK8, MAPK9, PAK4
Viral carcinogenesis	MAPK1, CCND1, CDK4, MAPK3, PIK3CA, CDK2, HDAC11, HDAC3, HDAC6, HDAC1, HDAC8, SRC, SYK, PIK3CD, CDK1, HDAC2
Yersinia infection	MAPK1, MAPK14, MAPK3, PIK3CA, MAP2K1, GSK3B, PTK2, MAPK10, MAP3K7, SRC, PIK3CD, RPS6KA3, MAPK8, MAPK9

**Table 2 molecules-27-08948-t002:** Interaction strength between 22-(4′py)-JA and potential targets.

Targets	PDB	Ligand Efficiency (kcal/mol per Heavy Atom)	Binding Energy (kcal/mol)	Interacting Position of Targets			
				H-Bonding	van der Waals	Hydro-Phobic	Electro-Static
ERK1 (MAPK3)	6GES	−0.225	−9.9	ASN171	ILE48, GLY49, GLU50, GLY51, LYS71, GLN122, LEU124, MET125, LYS168, PRO169, ASN171, LEU172	VAL56 ALA69 ILE101 TYR130 LEU173 CYS183	ASP128
ERK2 (MAPK1)	1WZY	−0.2	−8.8	LYS55 GLU71 ILE103 GLU109 LYS114 SER153	ALA52, ILE53, ILE56, LEU75, ILE84, LEU107, THR110, LEU156, ASN154	ILE31 VAL39 CYS166	ASP111
MEK1 (MAP2K1)	7PQV	−0.207	−9.1	GLN46 TYR125 GLU203 PHE371 ALA372	ARG49, LYS168, ILE171, ALA172, LYS175, GLY202, VAL369ASP370	PRO124	ASP43

## Data Availability

All data are available in the manuscript.

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
