# Peer review of "Target Identification of 22-(4-Pyridinecarbonyl) Jorunnamycin A, a Tetrahydroisoquinoline Derivative from the Sponge Xestospongia sp., in Mediating Non-Small-Cell Lung Cancer Cell Apoptosis"

_molecules, 2022, doi:10.3390/molecules27248948_

Round 1
Reviewer 1 Report (New Reviewer)
The manuscript has described that semisynthetic derivative 22-(4'py)-JA exhibited a potent inducing non-small-cell lung cancer (NSCLC)
apoptosis acitivty . Furthermore, ERK1/2 and MEK1 were identified as molecular targets of 22-(4'py)-JA in mediating
NSCLC apoptosis combining a cell-based assay and bioinformatic analysis. The research work is significant. The experiments are designed reasonable, and the results are promising for the clinical application against NSCLC. However, general speaking, the title should be as succinct as possible, I would suggest the authors delete "Thai blue" in the title. Another suggestion it is better for structure-activity relationship analysis to semisynthetize more derivatives for similar work in the future.
Author Response
Response: We are grateful for the reviewer’s very helpful comments. We have amended the manuscript according to the reviewer’s suggestion carefully. We have removed "Thai blue" from the title, and mentioned the structure-activity relationship analysis for further study in the discussion part (line 361-363). The changes were made in red. Our point-by-point responses are below. We are appreciated further revision if some points remained required for more clarification.

Reviewer 2 Report (New Reviewer)
The author reports the identification of a target for non-small cell lung cancer apoptosis by 22-(4-pyridinecarbonyl) jorunnamycin A. Author has provided a good introduction about Thai blue sponge Xestospongia sp. and the role of its active chemical constituents against cancer. Then the author presented a synthetic scheme followed by target identification and validation by in vitro cellular assay. The paper presents interesting results. But some important information is missing. Which should be addressed before publication.
1) In the introduction, the author has described different chemical modifications of jorunnamycin A and its activity. It would be great if the author provides the structure of those chemical modifications to make it easy for readers to understand.
2) In the synthetic scheme, the author has shown the use of EDCI for ester formation using acid chloride. Just curious to know if an author has tried reaction without EDCI. The acid chloride is very reactive and it’s easy to form ester just by using base.
3) The author should provide Proton NMR spectra with integrations.
4) NMR spectra for synthesized compound show lots of impurities present. I would encourage the author to include mass spec and percent purity (HPLC) data.
5) In Figure 7, the IC50 curves look great. But it would be great if the author includes a small table with IC50 values.
Author Response
Please see attached file for the revised Figure.
The author reports the identification of a target for non-small cell lung cancer apoptosis by 22-(4-pyridinecarbonyl) jorunnamycin A. Author has provided a good introduction about Thai blue sponge Xestospongia sp. and the role of its active chemical constituents against cancer. Then the author presented a synthetic scheme followed by target identification and validation by in vitro cellular assay. The paper presents interesting results. But some important information is missing. Which should be addressed before publication.
Response: We are grateful for the reviewer’s very helpful comments. We have amended the manuscript according to the reviewer’s suggestion carefully. The changes were made in red. Our point-by-point responses are below. We are appreciated further revision if some points remained required for more clarification.
1) In the introduction, the author has described different chemical modifications of jorunnamycin A and its activity. It would be great if the author provides the structure of those chemical modifications to make it easy for readers to understand.
Response: We are grateful for the reviewer’s very helpful comments. We add the chemical structure of different chemical modifications of jorunnamycin A in Figure 1 (Line 84).
Figure 1 22-O-Ester Derivatives of Jorunnamycin A [23]
2) In the synthetic scheme, the author has shown the use of EDCI for ester formation using acid chloride. Just curious to know if an author has tried reaction without EDCI. The acid chloride is very reactive and it’s easy to form ester just by using base.
Response: The alternative synthesis of 22-(4¢py)-JA regarding esterification between jorunnamycin A and isonicotinoyl chloride was involved the use of triethylamine as base and pyridine as solvent at -17 °C for 3 h to obtain 22-(4¢py)-JA with 56.8% yield (Charupant et al., 2009). The reaction without EDCI provides less product. Therefore, in this work, Steglich esterification was performed to increase the yield to 65%, which is a mild and neutral condition.
References
Charupant K, Daikuhara N, Saito E, Amnuoypol S, Suwanborirux K, Owa T, Saito N. Chemistry of Renieramycins. Part 8: Synthesis and Cytotoxicity Evaluation of Renieramycin M-Jorunnamycin A Analogues. Bioorg Med Chem. 2009 Jul 1;17(13):4548-58.
3) The author should provide Proton NMR spectra with integrations.
Response: Proton NMR spectra with integrations was included in supporting information (Figure S1 and Figure S2).
4) NMR spectra for synthesized compound show lots of impurities present. I would encourage the author to include mass spec and percent purity (HPLC) data.
Response: In the same batch, NMR spectra was performed again with higher concentration of compound (12 mg) and were included in the revised supporting information (Figure S1 and Figure S2). The result shown that the 1H-NMR spectra have impurities only at chemical shift around 1.24 and 0.85 ppm, as well as the chemical shift at 14.1, 22.0 and 31.0 of those of 13C-NMR, which correspond to the solvent peak (hexane). Moreover, we performed LC/MS chromatographic analysis to obtain higher resolution of mass data, and the of 22-(4¢py)-JA structure was confirmed by MS-MS experiment as shown in the revised supporting information (Figure S3). In addition, the purity of 22-(4¢py)-JA was evaluated by LC/MS chromatography which is 88.9% of purity. This data was included in the revised supporting information (Figure S4). We have mentioned them in line 104-105 of the revised MS.
5) In Figure 7, the IC50 curves look great. But it would be great if the author includes a small table with IC50 values.
Response: We have included IC50 in the revised Fig 8A (line 275).

Reviewer 3 Report (New Reviewer)
The authors investigated the derivative of Jorunnamycin A and its potential molecular target using multi-platform of Omics and in silico model prediction. They examined potential biomarkers after simulation of those algorithms represented in silico models such as protein interaction, etc.
They have highlighted different modes of action in Lung apoptosis using three different cell lines. For example, the phoisphoratuon of MEK and ERK molecules is expected to reduce anti-apoptotic molecule bcl-2. However, they need to be validated by functional analysis (e.g., siRNA or KO system) and caspase activity using in vitro assay to be confirmed the signaling pathway compared to JA and transformants such as 22-(4’py)-JA
Overall, the results they provide look solid and promising to achieve future goals. The length of the article is adequate, neatly documented clearly, and well justified their findings regarding objective-driven justification using their strategy. It looks like an interesting anti-lung cancer alternative which the authors may be challenged with animal studies in the near future.
It is a very interesting progress report though, that needs to be elaborated on signaling details along with the molecular network.
Minor issue
In the line 464 please check the typo (DMEM?)
Author Response
Response: We are grateful for the reviewer’s very helpful comments. We have amended the manuscript according to the reviewer’s suggestion carefully. The changes were made in red. Our point-by-point responses are below. We are appreciated further revision if some points remained required for more clarification. We have included the caspase experiment in Figure S7. The result demonstrated that 22-(4¢py)-JA incleased cleave-caspase 3, while it was not affected by JA. This data indicated that 22-(4¢py)-JA induced apoptosis via caspase 3-dependent mechanism, which is stronger than JA.
We agreed with the reviewer’s suggestion that the functional analysis is required to prove the signaling pathway. However, it is well-known that ERK/MEK plays a vital role for cell survival and proliferation, the inhibition of ERK by either knockdown or knockout experiment spontaneously induces cell death, therefore the treatment with 22-(4¢py)-JA in ERK/MEK depleted cells could not prove the effect of 22-(4¢py)-JA-mediating cell death via this signaling. To overcome this concern, we treated the cells with epidermal growth factor (EGF) in combination with 22-(4'py)-JA, whether MEK/ERK activated by EGF was suppressed by 22-(4'py)-JA, consequently decreasing viable cells. Since EGF, a ligand of receptor tyrosine kinase, is widely reported to promote tumor growth and cell proliferation, and suppress apoptosis through activation of its downstream signaling including MEK/ERK (Wee and Wang, 2017), combination treatment with EGF is more appropriate to provide precise mechanism of action of drug or compound targeting at MEK/ERK signaling (Dyshlovoy et al., 2012; Kawiak et al., 2019). Figure 9C demonstrated that 22-(4'py)-JA remarkedly decreased a number of viable cells in the EGF-treated group. EGF mediated an increase of MEK and ERK phosphorylation and Bcl-2 expression, in which they were reversely reduced in the presence of 22-(4'py)-JA significantly (Figure 9D). These data indicated that MAPK/Bcl-2 signaling is a mode of action of 22-(4'py)-JA in NSCLC.
Minor issue
In the line 464 please check the typo (DMEM?)
Response: We apologize for the unclear term used. The TMN buffer is the lysis buffer used for total protein extraction in the immunoblotting experiment. We have included the component in line 472-474 of the revised MS.
References
Dyshlovoy SA, Fedorov SN, Kalinovsky AI, Shubina LK, Bokemeyer C, Stonik VA, Honecker F. Mycalamide A Shows Cytotoxic Properties and Prevents EGF-Induced Neoplastic Transformation through Inhibition of Nuclear Factors. Marine Drugs. 2012; 10(6):1212-1224.
Kawiak A, Domachowska A, Krolicka A, Smolarska M, Lojkowska E. 3-Chloroplumbagin Induces Cell Death in Breast Cancer Cells Through MAPK-Mediated Mcl-1 Inhibition. Front Pharmacol. 2019;10:784.
Wee P, Wang Z. Epidermal growth factor receptor cell proliferation signaling pathways. Cancers (Basel). 2017;9(5):52.

This manuscript is a resubmission of an earlier submission. The following is a list of the peer review reports and author responses from that submission.
Round 1
Reviewer 1 Report
Well-organized study including in vitro and in silico experiments, as well as semi-synthesis part.
I would like to mention the following:
1) I believe that the authors should add an in vitro experiment where normal cells of peripheral blood, like lymphocytes, will be checked by MTT using the 22-(4’py)-JA derivative. It is very important to know if the molecule can kill normal cells in parallel with the cancer cell lines.
2) Moreover, I would like to see data that could possibly show in more details the apoptotic event. For example, if the cancer cells become early apoptotic or late apoptotic by the derivative .
Author Response
Manuscript ID# marinedrugs-1967440
Review#1
Well-organized study including in vitro and in silico experiments, as well as semi-synthesis part.
Response: We are grateful for the reviewer’s very helpful comments. We have amended the manuscript according to the reviewer’s suggestion carefully. The changes were made in red. Our point-by-point responses are below. We are appreciated for further revision if some points remained required for more clarification.
I would like to mention the following:
1) I believe that the authors should add an in vitro experiment where normal cells of peripheral blood, like lymphocytes, will be checked by MTT using the 22-(4’py)-JA derivative. It is very important to know if the molecule can kill normal cells in parallel with the cancer cell lines.
Response: We would like to thank the reviewer for raising a very important suggestion. We have included the additional cytotoxic experiment performing in normal lung BEAS-2B cells, since the compound targets at tumor in lung, which normal lung epithelial cells are located adjacently. The data shows that 22-(4’py)-JA has less cytotoxic to BEAS-2B cells with IC50 ³ 100 nM, whereas it has IC50 ~ 14-19 nM in cancer cell lines. It suggests that this compound induces cell death preferentially in lung cancer cells. We included this data in Fig. S4 and mentioned in the Result part.
2) Moreover, I would like to see data that could possibly show in more details the apoptotic event. For example, if the cancer cells become early apoptotic or late apoptotic by the derivative.
Response: We have included additional experiments according to the reviewer’s suggestion. Apoptosis cell death was confirmed by Annexin-V/propidium iodide (PI) staining. The data demonstrated that 22-(4'py)-JA mediated an increase number of early apoptotic and late apoptotic in a dose-dependent manner in all tested cells, confirming apoptosis inducing effect of 22-(4'py)-JA. The data were included in Figure S5 in the revised MS.

Reviewer 2 Report
In this manuscript, Iksen, et al. identified the potential target of 22-(4'py)-JA, a tetrahydroisoquinoline derivative from the Thai blue sponge Xestospongia sp. using computational-based analysis relying on network pharmacology and molecular docking approaches. Furthermore, the cytotoxic effect and molecular mechanism of 22-(4'py)-JA were verified by using an in vitro cell culture-based experiment. The research is interesting and would be useful for better understanding of the anti-tumor mechanism of 22-(4'py)-JA in NSCLC. However, the reviewer considers that the current study is much too preliminary, thus it is not suitable for publication on Marine Drugs.
Some Major Points:
1. Using molecular docking and molecular dynamic simulations to validate the drug-target interaction is insufficient to support the conclusion. CETSA or kinase inhibition assays would provide more convinced evidence.
2. The mechanism study is too superficial, MAPK signaling pathway inhibitor or siRNA should be used to demonstrate 22-(4'py)-JA induces apoptosis and inhibit NSCLC cell proliferation through inhibition of MEK-ERK pathway.
Author Response
Review#2
In this manuscript, Iksen, et al. identified the potential target of 22-(4'py)-JA, a tetrahydroisoquinoline derivative from the Thai blue sponge Xestospongia sp. using computational-based analysis relying on network pharmacology and molecular docking approaches. Furthermore, the cytotoxic effect and molecular mechanism of 22-(4'py)-JA were verified by using an in vitro cell culture-based experiment. The research is interesting and would be useful for better understanding of the anti-tumor mechanism of 22-(4'py)-JA in NSCLC. However, the reviewer considers that the current study is much too preliminary, thus it is not suitable for publication on Marine Drugs.
Response: We are grateful for the reviewer’s very helpful comments We have performed substantial in vitro experiments for verifying the molecular mechanism in revised manuscript to strengthen our finding, according to the reviewers’s suggestion. The changes were made in red. Our point-by-point responses are below. We are appreciated for further revision if some points remained required for more clarification.
Some Major Points:
- Using molecular docking and molecular dynamic simulations to validate the drug-target interaction is insufficient to support the conclusion. CETSA or kinase inhibition assays would provide more convinced evidence.
Response: We would like to thank the reviewer for raising a very important suggestion. We appreciate the reviewer concern. This work mainly focuses on the semi-synthesis of 22-(4'py)-JA derivative from marine organism and an identification of targets of new compound using network pharmacology approach, emphasizing the benefit of this approach as an intriguing model for target identification of new drugs. Therefore, we performed substantial in silico network pharmacology and molecular docking/dynamic experiments, and further additionally verified the target by in vitro cell culture-based assays. We totally agree that kinase inhibition assay provides supportive evidence confirming mode of action, but this experiment is beyond the scope of main purpose. Together with, due to the time constraints and available resource, we cannot conduct this experiment at this time. However, we performed additional experiments as the reviewer’s suggestion (comment no.2). This could sufficiently strengthen the molecular mechanism of 22-(4'py)-JA on MEK/ERK signaling. We would like to thank the reviewer again for providing the new insight into the development of this compound as MEK/ERK inhibitor. We would like to keep this experiment for further study involving with the modification of 22-(4'py)-JA for having more potency on MEK/ERK activity. This was included in the Discuss part of the revised MS.
- The mechanism study is too superficial, MAPK signaling pathway inhibitor or siRNA should be used to demonstrate 22-(4'py)-JA induces apoptosis and inhibit NSCLC cell proliferation through inhibition of MEK-ERK pathway.
Response: We would like to thank the reviewer for this comment. To confirm the mechanism of action on MEK activity, we treated the cells with epidermal growth factor (EGF) in combination with 22-(4'py)-JA instead, whether MEK/ERK activated by EGF was suppressed by 22-(4'py)-JA, consequently decreasing viable cells. Since EGF, a ligand of receptor tyrosine kinase, is widely reported to promote tumor growth, cell proliferation and apoptosis deregulation through activation of its downstream signaling including MEK/ERK (Wee and Wang, 2017), combination treatment with EGF is more appropriate to provide precise mechanism of action of drug or compound targeting at MEK/ERK signaling (Dyshlovoy et al., 2012; Kawiak et al., 2019). The results demonstrated that 22-(4'py)-JA remarkedly decreased a number of viable cells in EGF-treated group (Fig. S6A). Since A549 cells has been reported to have high aggressive cancer behaviors both in vitro and in vivo models (Shindo-Okada et al., 2002; Lacret et al., 2022), the mechanism of 22-(4’py)-JA on ERK/MEK signaling was primarily evaluated in this cell type. It showed that EGF mediated an increase of MEK and ERK phosphorylation and Bcl-2 expression, in which they were reversely reduced in the presence of 22-(4'py)-JA significantly (Fig. S6B). These data indicated that MAPK/Bcl-2 signaling is a mode of action of 22-(4'py)-JA in NSCLC. This data was included in Figure S6 in the revised MS.

Round 2
Reviewer 1 Report
The authors addressed appropriately almost every point I raised.
They used a normal lung cell line instead of the lymphocytes I had suggested. It was certainly a very positive addition to the strengthening of the manuscript.
However, I would also expect to see the effect of the molecule on peripheral blood cells, since in order to reach the site of the tumor if it is ever administered to a patient, it will be through the blood circulation. Moreover, it is known the myelotoxicity of most anticancer drugs.
Maybe, they could make a comment about it and study it in the future.
Reviewer 2 Report
Although the authors have supplemented some additional experiments, aiming to address the concern raised by the reviewer. However, the key problem (whether the drug target of 22-(4'py)-JA is MEK or ERK kinase?) still remain unclear. It is well known that EGF can also activate PI3K/AKT/mTOR signaling pathway, then induce activation of MEK/ERK signaling pathway. Therefore, these supplementary data cannot demonstrate 22-(4'py)-JA directly inhibit MEK/ERK kinase activity to induce apoptosis and inhibit NSCLC cell proliferation. Therefore, the reviewer also considers this paper should be rejected.